# The hierarchical organization of autocatalytic reaction networks and its relevance to the origin of life

**Zhen Peng**[1], **Jeff Linderoth**[1,2], **David A. Baum**[1,3]*

1 Wisconsin Institute for Discovery, University of Wisconsin-Madison, Madison, Wisconsin, United States of America, 2 Department of Industrial and Systems Engineering, University of Wisconsin-Madison, Madison Wisconsin, United States of America, 3 Department of Botany, University of Wisconsin-Madison, Madison, Wisconsin, United States of America

* dbaum@wisc.edu

**Data Availability Statement:** All relevant data are within the manuscript and its Supporting Information files.

**Funding:** This work was supported by grants to DAB from NASA (80NSSC17K0296) and the University of Wisconsin Vice-Chancellor for

## Abstract

Prior work on abiogenesis, the emergence of life from non-life, suggests that it requires chemical reaction networks that contain self-amplifying motifs, namely, autocatalytic cores. However, little is known about how the presence of multiple autocatalytic cores might allow for the gradual accretion of complexity on the path to life. To explore this problem, we develop the concept of a seed-dependent autocatalytic system (SDAS), which is a subnetwork that can autocatalytically self-maintain given a flux of food, but cannot be initiated by food alone. Rather, initiation of SDASs requires the transient introduction of chemical "seeds." We show that, depending on the topological relationship of SDASs in a chemical reaction network, a food-driven system can accrete complexity in a historically contingent manner, governed by rare seeding events. We develop new algorithms for detecting and analyzing SDASs in chemical reaction databases and describe parallels between multi-SDAS networks and biological ecosystems. Applying our algorithms to both an abiotic reaction network and a biochemical one, each driven by a set of simple food chemicals, we detect SDASs that are organized as trophic tiers, of which the higher tier can be seeded by relatively simple chemicals if the lower tier is already activated. This indicates that sequential activation of trophically organized SDASs by seed chemicals that are not much more complex than what already exist could be a mechanism of gradual complexification from relatively simple abiotic reactions to more complex life-like systems. Interestingly, in both reaction networks, higher-tier SDASs include chemicals that might alter emergent features of chemical systems and could serve as early targets of selection. Our analysis provides computational tools for analyzing very large chemical/biochemical reaction networks and suggests new approaches to studying abiogenesis in the lab.

## Author summary

The level of complexity seen in even the simplest living system is too great to have arisen in its current form without a long history of complexification. In this paper, we explore

Research and Graduate Education. ZP and DAB received salary support from the NASA grant and ZP also received salary support from the University of Wisconsin. The funders had no role in study design, data collection and analysis, decision to publish, or preparation of the manuscript.

**Competing interests:** The authors have declared that no competing interests exist.

the view that open environments on the early Earth that received an ongoing flux of food chemicals could have complexified gradually by the sequential activation of autocatalytic chemical reaction systems. We develop the concept of seed-dependent autocatalytic systems (SDASs)–subnetworks whose components can self-propagate once activated by "seed" molecules, which might result from rare reactions or import from other environments. We developed new computational tools for detecting SDASs in reaction databases and determining if they are hierarchically organized, such that the activation of a lower-tier SDAS allows a higher-tier SDAS to then be seeded, much like the relationship between producers and consumers in an ecosystem. We apply our algorithms to two chemical reaction networks, one biological and the other abiotic, and find that both contain hierarchically organized SDASs. These results support the fundamental continuity of the way that the chemistry of non-life and life is organized and suggest new classes of laboratory experiment.

## 1. Introduction

The core puzzle of abiogenesis is, given a flux of energy and simple materials as food (e.g., water, carbon dioxide, and minerals), what system could arise spontaneously with the capacity of self-propagation and adaptive complexification. As a result, any successful theory of abiogenesis needs to specify a system, a "first evolver," that is endowed with the capacity to evolve adaptively and accrete complexity yet is simple enough to have a reasonable probability of arising spontaneously on the prebiotic Earth.

A number of prior investigations into the origin of life have started by defining the key components of cellular life and imagining that the first evolver had simpler versions of these components [1–6]. While these models have utility for abstracting key properties of life, it is unlikely that multiple interdependent complex modules, such as genetic polymers and selectively permeable membranes, could arise in a coordinated manner by chance alone [7,8]. The idea that the first evolver was an RNA or a set of RNAs that performed the functions of an RNA-dependent RNA polymerase [9–11] is appealing, but has several well-known problems, including the challenge of explaining the occurrence of a driving flux with high-enough concentrations of activated β-D-ribonucleotides without prior autocatalysis and evolution [12,13]. Rather, we subscribe to the metabolism-first view that evolvability predated the origin of genetic polymers and arose from the dynamics of chemical reaction networks [14,15].

Whatever the nature of the first evolver, it must have been able to self-propagate because a system that lacks a way to make more of itself has no way to generate descendants that can manifest heritable differences, as is required for evolution. Moreover, self-propagation ability is needed for a system to maintain status quo in any open environment, since dilution and other disturbances are inevitable in the long run. As a result, for the first evolver to exhibit two core attributes of life, namely self-maintenance and the capacity to evolve [16], it must have been autocatalytic [17,18]. A core challenge for origin-of-life models, therefore, is to explain how, from its earliest inception and through all subsequent evolutionary innovations, proto-biological systems in open environments could autocatalytically convert replenishing food into more internal components. As discussed further in Sect. 3.3, Chemical Organization Theory (COT), which is an otherwise promising candidate for explaining abiogenesis [19,20], does not necessarily require autocatalysis because it does not enforce environmental openness. Therefore, COT, in its current form, is not optimal for elucidating abiogenesis.

The theories of collectively autocatalytic sets (CASs) [21], and reflexively autocatalytic, food-generated sets (RAFs) [22,23], have been used extensively to explore autocatalysis in relation to abiogenesis [7,24]. Both models assign a central role to *explicit catalysis*: for a chemical reaction network represented by connected nodes of species and reactions, explicit catalysis applies when a reaction node is directly catalyzed by one or multiple species nodes within the network. Although explicit catalysis is highly enriched in modern metabolism, primarily due to the activity of enzymes and ribozymes, we believe that it is improbable that most reactions in the first evolver were explicitly catalyzed by internally synthesized catalysts [25]. Considering the relative simplicity of the prebiotic Earth, it was more likely that most reactions in the first evolver occurred without explicit catalysis or depended on simple environmental catalysts. Moreover, an emphasis on explicit catalysis can distract attention from a key feature of autocatalytic systems, namely the potential for stoichiometric increase of the system's internal components [17]. An autocatalytic cycle can show stoichiometric autocatalysis even if it does not include any explicit catalysts [17,26]. For example, with A provided as food, $A + B \rightleftharpoons C$, $C + A \rightleftharpoons 2B$ is an autocatalytic cycle but not a RAF. Equally, one can identify networks that are formally RAFs or pseudo-RAFs [27] but are not stoichiometrically autocatalytic. For example, a linear chain of explicitly catalyzed reversible reactions, $A \overset{U}{\rightleftharpoons} B \overset{V}{\rightleftharpoons} C$, where the catalysts U and V are in the food set, forms a RAF if A, B, or C is in the food set, or a pseudo-RAF if A, B, and C are not in the food set. However, this chain of reactions lacks the potential for stoichiometric increase. For these reasons, it is desirable to develop a theory of abiogenesis that focuses on stoichiometric autocatalysis rather than explicit catalysis, while still allowing for cases of explicit catalysis on single reactions as well as catalysis based on a series of reactions over which a chemical species is first consumed and then fully regenerated.

Being autocatalytic is necessary but not sufficient for evolution. There has been much discussion about what additional ingredients are needed for evolvability [28–31], but several scientists have suspected that the accretion of complexity is possible when chemical reaction systems contain multiple interacting autocatalytic motifs that can be triggered sequentially by rare, stochastic events [26,32–34]. However, we still lack a clear understanding of how stoichiometrically autocatalytic motifs are organized in real chemical reaction networks and whether that organization allows for gradual, adaptive complexification.

We develop, here, the concept of a seed-dependent autocatalytic system (SDAS), a stoichiometrically autocatalytic subnetwork that can be activated by introduction of a chemical "seed" and then sustain itself even in the absence of the exogenous seed. First, we use a hypothetical reaction network to introduce the SDAS concept and explain how it could enable stepwise complexification due to the fact that seeding of one SDAS can result in the sustained production of more species, some of which could serve as food for additional, higher-tier SDASs. Then we propose an algorithm, based on network expansion, stoichiometry, and linear programming, to identify and analyze SDASs in chemical reaction databases. Finally, we use the algorithm to compare two databases of real reactions: an abiotic database including radiolytic and geochemical reactions [35], and a subset of the KEGG biochemical database that putatively represents a primordial metabolic network [36]. We find similar topological features and trophic hierarchy in the two networks, but also some interesting differences that may be signatures of evolution following the appearance of genetically encoded enzymes. Our results support the idea that sequential activation of SDASs, combined with occasional deactivation of previously active SDASs, may be a primordial mechanism of evolution before the origin of genetic polymers. We end by discussing implications of our results, providing brief comparisons of COT, RAF theory, and SDAS theory, and discussing experiments that could be conducted to test the SDAS theory of abiogenesis.

## 2. Results

### 2.1. Complexification of reaction networks based on seed-dependent autocatalytic systems

We will start by providing several examples to introduce the key concepts informally, with definitions provided in Table 1.

Autocatalysis usually refers to the phenomenon that a process generates a product that catalyzes the process, whether by acting as an explicit catalyst or by otherwise accelerating the rate at which the reaction proceeds. For instance, A + F → 2A is an autocatalytic reaction, which can also be written to emphasize the catalytic effect as: F $\xrightarrow{A}$ A. In an open environment where F constantly flows in but A cannot spontaneously emerge, A + F → 2A is a simple SDAS because a *seed* comprising the member species, A, is able to trigger the sustainable conversion of F into A. Here, A is a *supported seed*. Alternatively, this SDAS could be seeded by a species that can be converted into A; for example, given another reaction B ⇌ A + C, B is an *unsupported seed*, because sustained production of B also requires C, which is not provided as environmental food. However the SDAS is triggered, it is not guaranteed to be able to self-maintain in practice, because the reaction(s) may not be fast enough to overcome ongoing dilution [26]. This illustrates that SDASs are topological not kinetic features of chemical reaction networks.

A seed, supported or unsupported, could comprise a single chemical species (e.g., A or B in the preceding), being a *singleton seed*, or it could comprise a set of chemicals that need to be seeded together to trigger a SDAS, being a *composite seed*. For example, B and C would be a composite seed for the SDAS: B + C ⇌ A, A + F → 2A (with F provided as food).

Finally, it is possible for the same SDAS to be triggered by multiple singleton seeds, which we will call a *clique*. For example, for the SDAS composed of these two reactions, A + F ⇌ B, B + F → 2A (with F provided as food), A and B belong to the same clique because either is sufficient to induce the SDAS. In this paper, we only consider cliques with supported seeds, because unsupported seeds may induce the same SDAS but different non-SDAS reactions; for example, unsupported seeds B and D might induce A + F → 2A via different reactions B ⇌ A + C and D ⇌ A + E.

To illustrate the way that chemical reaction networks with SDASs might enable evolution-like dynamics, we have developed a hypothetical network that includes multiple SDASs

**Table 1. Definitions of key concepts.**

| Concept | Definition |
|---|---|
| Network expansion operation | Given a database of allowed reactions, a starting set of chemical species, and a starting set of reactions, iteratively add new chemical reactions to the reaction set whenever all the reactants (and catalysts, if applicable) are present in the species set and then add all new products to the species set, until no more reactions can be added to the reaction set. |
| SDAS | A reaction network consisting of chemical species and reactions that cannot be activated by network expansion from food but, once activated, can be sustained by converting food into all its internal species. |
| Seed | A set of non-food chemical species that can induce a SDAS. |
| Singleton seed | A seed consisting of just one chemical species. |
| Composite seed | A seed set consisting of two or more chemical species that can jointly trigger a SDAS, with every proper subset of this seed set unable to trigger a SDAS. |
| Supported seed | A seed that is produced by the SDAS that it induces. |
| Unsupported seed | A seed that is not produced by the SDAS that it induces. |
| Clique | A set consisting of multiple singleton supported seeds that each triggers the same SDAS[a]. |

[a]For cliques containing composite seeds, please refer to Materials and methods (Sect. 4.3).

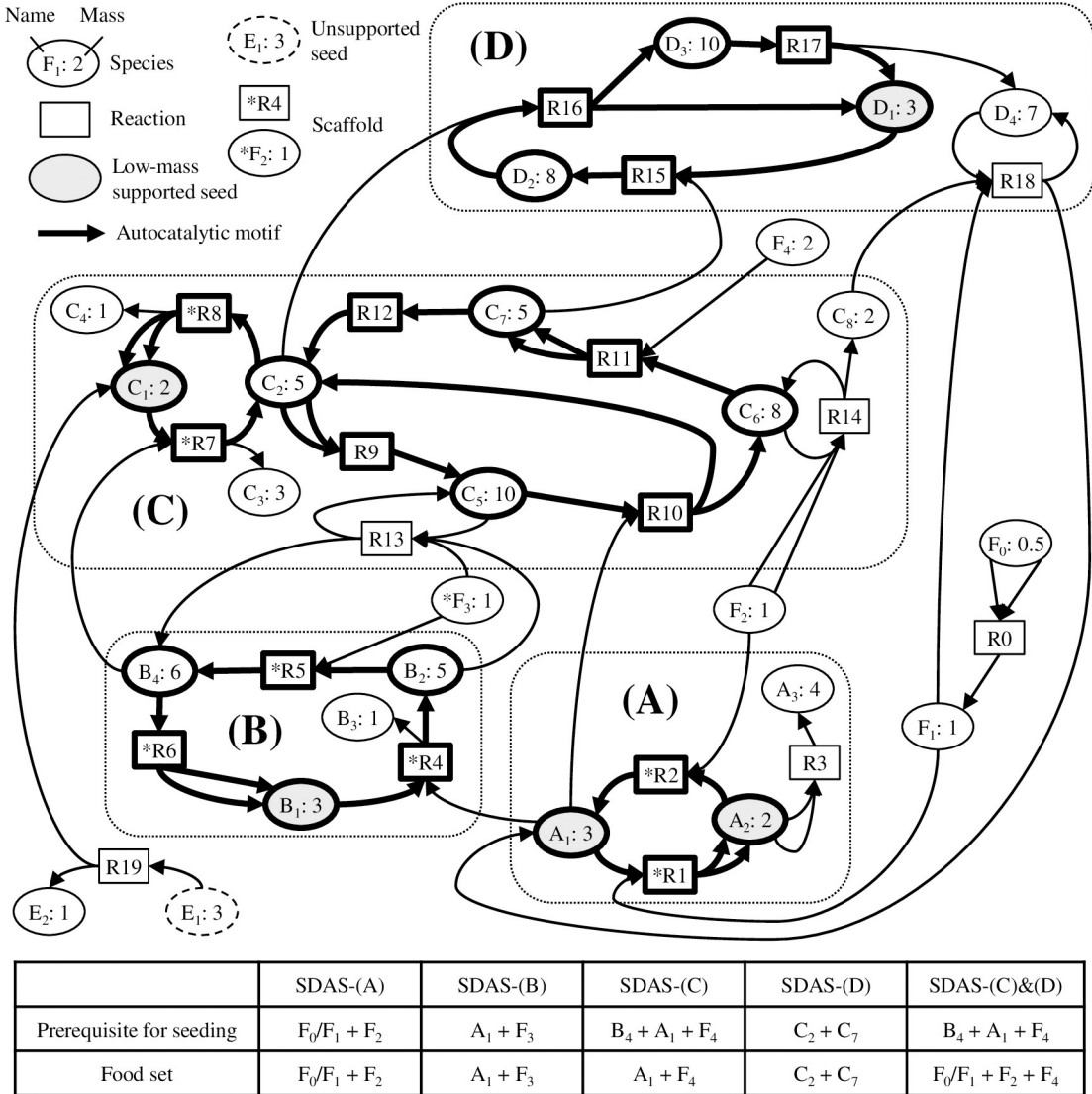

| | SDAS-(A) | SDAS-(B) | SDAS-(C) | SDAS-(D) | SDAS-(C)&(D) |
|---|---|---|---|---|---|
| Prerequisite for seeding | $F_0/F_1 + F_2$ | $A_1 + F_3$ | $B_4 + A_1 + F_4$ | $C_2 + C_7$ | $B_4 + A_1 + F_4$ |
| Food set | $F_0/F_1 + F_2$ | $A_1 + F_3$ | $A_1 + F_4$ | $C_2 + C_7$ | $F_0/F_1 + F_2 + F_4$ |

**Fig 1. A hypothetical example of a SDAS-organized reaction network.** Each species is associated with its mass, which can be viewed as a simple proxy for chemical complexity, where ultimate food and potential seeds all have mass $\leq 3$. This hypothetical example consists of four SDASs feeding on different sets of food. Each SDAS includes the member species of an autocatalytic motif and may also include species that are derived from the motif but are not part of the motif itself; for example, $A_3$ in SDAS-(A) and $B_3$ in SDAS-(B).

(Fig 1). Autocatalytic motifs, which may vary from a single reaction to a complex cycle of reactions involving multiple *member* species [17,26], are shown with bold arrows. SDASs may also include species that are not members of autocatalytic motifs themselves, as illustrated by $A_3$ (Fig 1A) and $B_3$ (Fig 1B). Some of these non-member species, such as $B_3$ (Fig 1B) and $D_4$ (Fig 1D), are byproducts of the reactions within autocatalytic motifs, and we call them *waste* species. A single SDAS can contain multiple distinct or intersecting autocatalytic motifs, the latter illustrated by SDAS-(C) (Fig 1).

An activated SDAS increases the diversity and perhaps complexity of the chemical species that can be sustained in an open environment. Although the best measurement of molecular complexity is yet to be determined [37–39], in this paper we will consider a molecule as more

complex if it has larger mass, more atoms and atom types, more bonds and bond types, and more rings. If we assume that the ultimate food species are simple, it is obvious that more reaction steps are generally needed to produce a more complex molecule [38]. Therefore, the maximum complexity of SDAS-supported species, relative to that of the SDAS-consumed food, is likely limited by SDAS size.

Some species of an initial SDAS could be essential for another SDAS to be sustained. In this case, it may be possible to seed the latter, higher-tier SDAS only after the lower-tier SDAS has been activated (Fig 1). This resembles a biological ecosystem where a higher-trophic-tier consumer can only invade an ecosystem that already contains suitable lower-trophic-tier organisms. Because higher-tier-SDAS members likely require more reactions to be synthesized from environmental food, on average we might expect higher chemical complexity in higher-tier SDASs than in lower-tier SDASs. Consequently, a reaction system receiving low-complexity food influx in an open environment may complexify by iterative seeding of sequentially higher-tier SDASs (Fig 1A–1D).

Sequential activation of SDASs may not only increase species diversity and complexity, but also induce emergent properties. For example, as the number of maintained species and reactions increases, the probability that at least some reactions are explicitly catalyzed by sustained species increases (e.g., $C_5$ catalyzes R13 in Fig 1). In addition, some physicochemical properties, such as amphiphilicity and chelation ability, may only be possible if species complexity exceeds some threshold. Thus, accretion of SDASs by sequential seeding might be associated with non-linear, emergent complexification.

An interesting potential effect of SDAS-organized networks is scaffolding. This applies when a lower-tier structure, a *scaffold*, helps build a higher-tier structure but, once built, the higher-tier structure is robust to the removal of the scaffold. For example, in Fig 1, SDAS-(B) scaffolds SDAS-(C) because SDAS-(B) is essential for activating SDAS-(C) but not for its maintenance. Scaffolding is also seen when higher-tier SDASs remove the need for lower-tier processes. For example, R14 in SDAS-(C) and R18 in SDAS-(D) jointly produce $A_1$, which initially required SDAS-(A) for its production (Fig 1). However, following activation of SDAS-(C) and SDAS-(D), SDAS-(A) ceases being essential for the production of $A_1$ or for persistence of the other SDASs. Thus, if SDAS-(A) were deactivated, for example due to an environmental change that rendered some reactions unfavorable, SDAS-(C) and SDAS-(D) could nonetheless persist.

To summarize this conceptual framework, sequential activation of hierarchically organized SDASs provides a potential mechanism for reaction networks to complexify in an open environment. Such a network architecture would mean that rare reactions or occasional influx of chemicals could induce stepwise complexification in an environment receiving simple species as food, as commonly assumed in abiogenesis scenarios [15,40–44]. Moreover, some higher-tier SDASs could alter emergent properties, for example conferring resilience to environmental perturbations, potentially mediating primordial adaptation. However, to validate this framework and examine whether it might enable the gradual acquisition of life-like properties, we need tools for detecting and analyzing SDASs in real reaction networks.

## 2.2. Detecting SDASs and analyzing intra- and inter-SDAS interactions

Real reaction databases are much more complicated than Fig 1, requiring computational methods to detect and analyze SDASs. Here, we developed methods based on network topology. While algorithms have been developed for detecting RAFs in reaction databases [36,45], our algorithm focuses on explicitly tracking stoichiometry, which is largely ignored by RAF theory but is the core attribute determining whether an autocatalytic system has the potential

to self-maintain in an open environment [17,26]. Compared to other algorithms considering stoichiometry [19,20], our methods feature environmental openness and the usage of seeding to separate internal and external subnetworks. Separating internal and external subnetworks by seeding and choosing food sets based on prior knowledge of prebiotic chemistry are key to detecting autocatalysis in reaction networks despite this being an NP-complete problem [46]. The algorithm we develop scales well, allowing for easy analysis of reaction networks that contain thousands of species and reactions.

To generate an organized subnetwork for analysis, we use a network expansion operation [47], $(S_E, R_E) = \Xi(S_O, R)$, which calculates all reactions $R_E$ and species $S_E$ that can be accessed given a starting set of species $S_O$ and a set of allowed reactions $R$ (see Materials and methods). A reaction network can be represented by a stoichiometric matrix, where rows represent species and columns represent reactions, with entries representing stoichiometric coefficients [17]. Usually, reactants have negative entries while products have positive ones. However, for explicitly catalyzed reactions, catalysts have non-negative entries even though they are required for the reaction, meaning that the corresponding columns need to be annotated to indicate dependence on explicit catalysis.

Let us define a set $S_U$ as *ultimate food*, which are assumed to be constantly supplied in an open environment. We also assume that the ultimate food species are generally simple and highly abundant, and that environmental openness makes every chemical species directly or indirectly subject to dilution. The set $S_0$ of species available as external food for a potential tier-1 SDAS is defined by $(S_0, R_0) = \Xi(S_U, R)$. We will call $(S_0, R_0)$ the tier-0 system (Figs 2 and S1A–S1D).

To search for a tier-1 SDAS feeding on $S_0$, we introduce a candidate supported seed $H$ containing one or multiple non-$S_0$ species, and calculate $(S_{C1}, R_{C1}) = \Xi(S_0 \cup H, R)$. We can say that $(S_1, R_1)$, where $S_1 = S_{C1} \setminus S_0$ and $R_1 = R_{C1} \setminus R_0$, is an internal subnetwork induced by $H$ (Figs 2 and S1E–S1H). The sufficient and necessary condition for $(S_1, R_1)$ to be a SDAS is that its corresponding rows (Fig 2, $\forall i \in [p, m]$) and columns (Fig 2, $\forall j \in [q, n]$) have the property that a vector of non-negative elements $x = (x_q, x_{q+1}, \ldots, x_n)$ exists such that [17]

$$\sum_{j=q}^{n} x_j s_{ij} > 0 \ (x_j \geq 0) \ \forall i \in [p, m], \tag{1}$$

where $s_{ij}$ is the entry at the $i$th row and $j$th column of the stoichiometric matrix. Inequality (1) means that there are some linear combinations of internal reactions such that all internal species can be produced with excess by consuming external species. When this holds, the internal subnetwork has the potential to compensate for the loss of internal species due to environmental openness. Linear programming is used to determine whether (1) can be satisfied [48] (see Materials and methods). If (1) is satisfied, we say that $H$ is a supported seed inducing a tier-1 SDAS $(S_1, R_1)$. For $H$ to be a composite supported seed, all elements of $H$ must be jointly necessary to induce the SDAS: $H$ must induce a SDAS and for any $H' \subset H$, $H'$ cannot induce a SDAS. For two singleton supported seeds to be in the same clique, they must induce the same SDAS.

An example in Fig 1 can help understand how the method works. Let us consider the reaction set $R$ that includes all reactions shown in the figure. Assuming that the ultimate food $S_U = \{F_0, F_2\}$, network expansion $\Xi(S_U, R)$ will make $S_0 = \{F_0, F_1, F_2\}$ and $R_0 = \{R0\}$. Suppose we try a candidate supported seed $H = \{A_1\}$. Network expansion $\Xi(S_0 \cup H, R)$ will make $S_{C1} = \{F_0, F_1, F_2, A_1, A_2, A_3\}$ and $R_{C1} = \{R0, R1, R2, R3\}$, which leads to $S_1 = \{A_1, A_2, A_3\}$ and $R_1 = \{R1, R2, R3\}$. Because the combination of 4 R1's, 5 R2's, and 1 R3 will form a net reaction $4F_1 + 5F_2 \rightarrow A_1 + A_2 + A_3$, inequality (1) is satisfied.

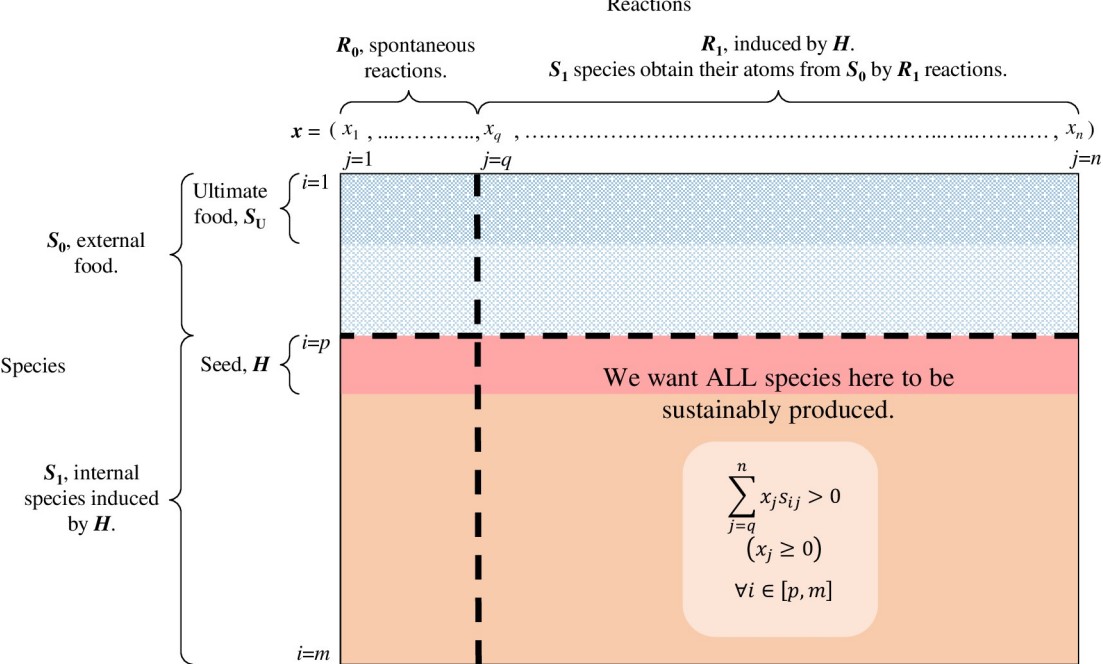

**Fig 2. Detecting a SDAS.** Based on network expansion, a stoichiometric matrix is divided into four segments. Network expansion from the ultimate food generates the upper left segment, consisting of external food available for a potential SDAS and reactions among these food species. Then adding the candidate seed triggers further network expansion, generating the other three segments. The lower left segment consists only of zeroes because the reactants and products of corresponding reactions are all external. The upper right segment represents the external food involved in the reactions induced by the seed; because external food is assumed freely available, detection of a SDAS does not rely on this segment. The lower right segment represents internal species and reactions induced by the seed; for these internal species to be produced with excess such that they have the potential to sustain dilution, the seed-induced reactions must find a way to synthesize all internal species by consuming external food.

To find all tier-1 SDASs inducible by a singleton supported seed, and organize them into cliques, we can simply scan all non-food species involved in $R$ one-by-one for supported seeds and determine whether they induce the same SDAS. One could also do the same for all pairs of species, all triplets, etc., although the search will likely become computationally prohibitive for composite seeds composed of many chemicals. To moderate the computational challenge, and to better reflect the idea that (almost) simultaneous seeding of multiple chemical species, especially complex ones, is rare, one may wish to focus more on lower-complexity singleton candidate seeds.

Although each SDAS must contain at least one autocatalytic motif, not every reaction or species in the SDAS is necessarily a member of an autocatalytic motif. Additionally, there might be multiple autocatalytic motifs within a single SDAS (as in Fig 1C). To identify autocatalytic motifs within SDASs, we developed an integer programming procedure that finds a vector, $x$, with a specified number of positive elements such that the internal species involved in the reactions corresponding to these positive elements are all sustainably produced (see Materials and methods). This procedure can be used to find autocatalytic motifs satisfying target criteria, such as containing the fewest reactions.

Once a SDAS is detected, all species in the SDAS can be treated as external food for higher-tier SDASs. For a multi-SDAS system, pairwise inter-SDAS interactions are rather like ecological symbioses between organisms [26]. SDASs may compete if they feed on common food; a higher-tier SDAS ($S_{high}$, $R_{high}$) may be a predator or parasite of a lower-tier SDAS ($S_{low}$, $R_{low}$) if ($S_{high}$, $R_{high}$) feeds on a member species of an autocatalytic motif within ($S_{low}$, $R_{low}$), or

SDASs may be mutualistic if ($S_{high}$, $R_{high}$) feeds on a waste product of an autocatalytic motif within ($S_{low}$, $R_{low}$). Additionally, SDASs may facilitate other ecological interactions, for example when ($S_{high}$, $R_{high}$) provides catalysts activating new paths linking key nodes in ($S_{low}$, $R_{low}$), or may confer broader ecosystem services, such as when ($S_{high}$, $R_{high}$) provides species with physicochemical properties protecting ($S_{low}$, $R_{low}$) against dilution. As with ecological interactions between organisms, pairs of SDASs may simultaneously engage in multiple direct and indirect interactions.

## 2.3. Abiotic and biochemical reaction databases

The abiotic reaction database that we analyzed was extracted from seven decades of published data [35], including free radical reactions, mineral geochemical reactions, amino acid production, chloride radical and polar reactions, nitrile radical and polar reactions, RNA nucleotide assembly, nuclear decay, and physicochemical reactions. To this database, we added a few reactions of the well-known abiotic formose reaction [49], but without formaldehyde dimerization because it is very slow and its reaction mechanism is yet to be determined [50,51].

The biochemical reaction database that we studied was obtained by removing reactions only present in eukaryotes and reactions dependent on $O_2$ [36] from the KEGG reaction database [52–54]. Xavier et al. claimed that this reaction database could be a proxy for a primordial metabolic network [36]. We added five spontaneous, reversible reactions that are missing from KEGG, for example, $H_2O \rightleftharpoons H^+ + OH^-$ and $H_2CO_3 \rightleftharpoons H^+ + HCO_3^-$, and one reaction (R06974) that entails a reaction mechanism very similar to a reaction (R06975) kept by Xavier et al. [36] (see Materials and methods; S2 Fig). We acknowledge that, in contrast to the abiotic network, most of the reactions in the KEGG database are catalyzed by enzymes. Therefore, it is likely that many of the biochemical reactions could not occur at high rates in a prebiotic world before biological catalysts evolved. Nonetheless, since all these reactions are chemically feasible, we reasoned that the relationships between reactants and products in such a curated biochemical reaction network is meaningful and that features shared by it and the abiotic network should be relevant to the origin of life. Thus, our goal is not to claim that the biochemical reactions in this database were exactly those that applied during the origin of life but to explore whether a network with this topology could permit stepwise complexification.

After preprocessing (see Materials and methods), the abiotic database consists of 277 species and 717 unidirectional reactions (S1 Table) and the biochemical database consists of 4216 species and 8402 reactions (S2 Table). Both databases exhibit highly right-skewed power-law histograms of the numbers of reactions that a species is involved in (S3 Fig), meaning that a randomly picked species is unlikely to be involved in many reactions.

For studies of the abiotic and biochemical databases, we chose {$H_2$, $CH_4$, NO, $FeS_2$, visible light} and {$H_2O$, $CO_2$, $NH_3$, $H_2S$, $H_2SO_4$, $H_2SO_3$, $HSO_3^-$, $H_3PO_4$, $H_4P_2O_7$} as the ultimate food, respectively. In addition, metals and minerals that might serve as catalysts are assumed freely available for biochemical reactions. These choices, although arbitrary, should be sufficient for validating the SDAS-based framework for cases starting with chemically simple food stocks.

## 2.4. Abiotic SDASs

The tier-0 system has 5 species and no reactions. Twelve of the 272 non-tier-0 species constitute the clique abio-1 inducing a tier-1 SDAS, SDAS-abio-1 (S3 Table) of 91 species and 220 reactions (S4 Table). The smallest autocatalytic motif within SDAS-abio-1 includes 10 reactions. There are 10 alternative (largely overlapping) 10-reaction autocatalytic motifs and at least 30 alternative 11-reaction autocatalytic motifs (S5B Table). As an example, one

10-reaction autocatalytic motif feeds on $CH_4$, NO, and visible light and produces $H_2CNH$ and infrared light as waste (Fig 3, S5A Table). For this motif, $H_2O$ is the explicit catalyst for reaction Rn387 (Fig 3).

The abiotic network includes examples of composite seeds. For example, neither the formyl radical (HCO) nor $O_2$ can induce a SDAS (S3 Table), yet together they can induce SDAS-abio-1 (S6 Table). Unsupported seeds are also present: 18 species are singleton unsupported seeds for SDAS-abio-1 (S7 Table). For example, the $C_2H_3$ radical is not a supported seed but causes the formation of OH radicals, which belongs to clique abio-1 (S1 and S3 Tables).

Once SDAS-abio-1 is activated and its species are available as food, 13 species constitute the clique abio-2 inducing a 14-species, 35-reaction tier-2 SDAS, SDAS-abio-2 (S8 and S9 Tables). SDAS-abio-2 includes the formose reaction, which feeds on $H_2CO$ synthesized by SDAS-abio-1 to produce monosaccharides. These sugars might modify the physicochemical properties of a local environment, such as viscosity or surface adsorption [55–58], potentially affecting the rate of loss of other species from the environment.

## 2.5. Biochemical SDASs

Network expansion from the ultimate food generates a 30-species, 44-reaction tier-0 system (S10 Table). Treating this tier-0 system as external food, 304 of the 4186 non-tier-0 species are singleton supported seeds able to induce a tier-1 SDAS (S11 Table). These seeds belong to three cliques: the 267-species clique bio-1a induces SDAS-bio-1a (301 species; 736 reactions; S12 Table); the 34-species clique bio-1b induces SDAS-bio-1b (357 species; 916 reaction; S13 Table); the 3-species clique bio-1c induces SDAS-bio-1c (1414 species; 4114 reaction; S14 Table). These SDASs are nested, with SDAS-bio-1c including the entirety of SDAS-bio-1b, which includes the entirety of SDAS-bio-1a. These SDASs share the same set of minimal autocatalytic motifs: there are 24 alternative 22-reaction autocatalytic motifs (S15B Table). One of them feeds on $H_2O$, $CO_2$, and $H_4P_2O_7$ and produces $H_3PO_4$ and $H_2O_2$ as waste (Fig 4, S15A Table). Within this motif, the carboxylation of pyruvic acid ($CH_3$-CO-COOH) to oxaloacetic acid (HOOC-$CH_2$-CO-COOH) by reaction R00217.b (Fig 4) is an important step. This same conversion may also be conducted in two steps via reactions R11074.b and R01447.b, meaning that (S)-lactic acid ((S)-$CH_3$-CHOH-COOH) would qualify as a catalyst if the composite reaction rate of R11074.b+R01447.b is higher than that of R00217.b (Fig 5). Also, it is worth noting that SDAS-bio-1a has the potential to facilitate catalysis by chelated metal ions because it contains multiple organic molecules that have chelating ability (e.g., citric acid) or are important precursors of chelating agents (e.g., glycine and aspartic acid).

Again, there are cases of composite seeds. For example, neither formaldehyde ($H_2CO$) nor acetic acid ($CH_3COOH$) is a seed (S11 Table), yet together they can induce SDAS-bio-1a (S16 Table). Also, there are 356 singleton unsupported seeds for SDAS-bio-1a (S17A Table), 145 for SDAS-bio-1b (S17B Table), and 1 for SDAS-bio-1c (S17C Table). For example, hypotaurine ($H_2N$-$CH_2$-$CH_2$-SOOH) is not produced by SDAS-bio-1a but can cause the formation of L-alanine ($CH_3$-CH($NH_2$)-COOH), which belongs to the bio-1a clique (S12 Table).

It is worth noting that the bio-1a clique has members as simple as acetylene ($C_2H_2$) and glycolaldehyde (CHO-$CH_2OH$). In contrast, every species in clique bio-1b is a pyrimidine nucleoside or a derivative thereof and contains at least 9 carbon atoms, and clique bio-1c consists of three species ($NAD^+$, NADH, and deamino-$NAD^+$) that each contains 21 carbon atoms. Considering that fortuitous introduction of a complex seed is unlikely, and treating molecular size as a crude proxy for complexity, we decided to explore whether the network underlying SDAS-bio-1b or SDAS-bio-1c could be activated after SDAS-bio-1a by simpler supported seeds. We found that after SDAS-bio-1a is activated, a 6-species clique (bio-2a), in which the

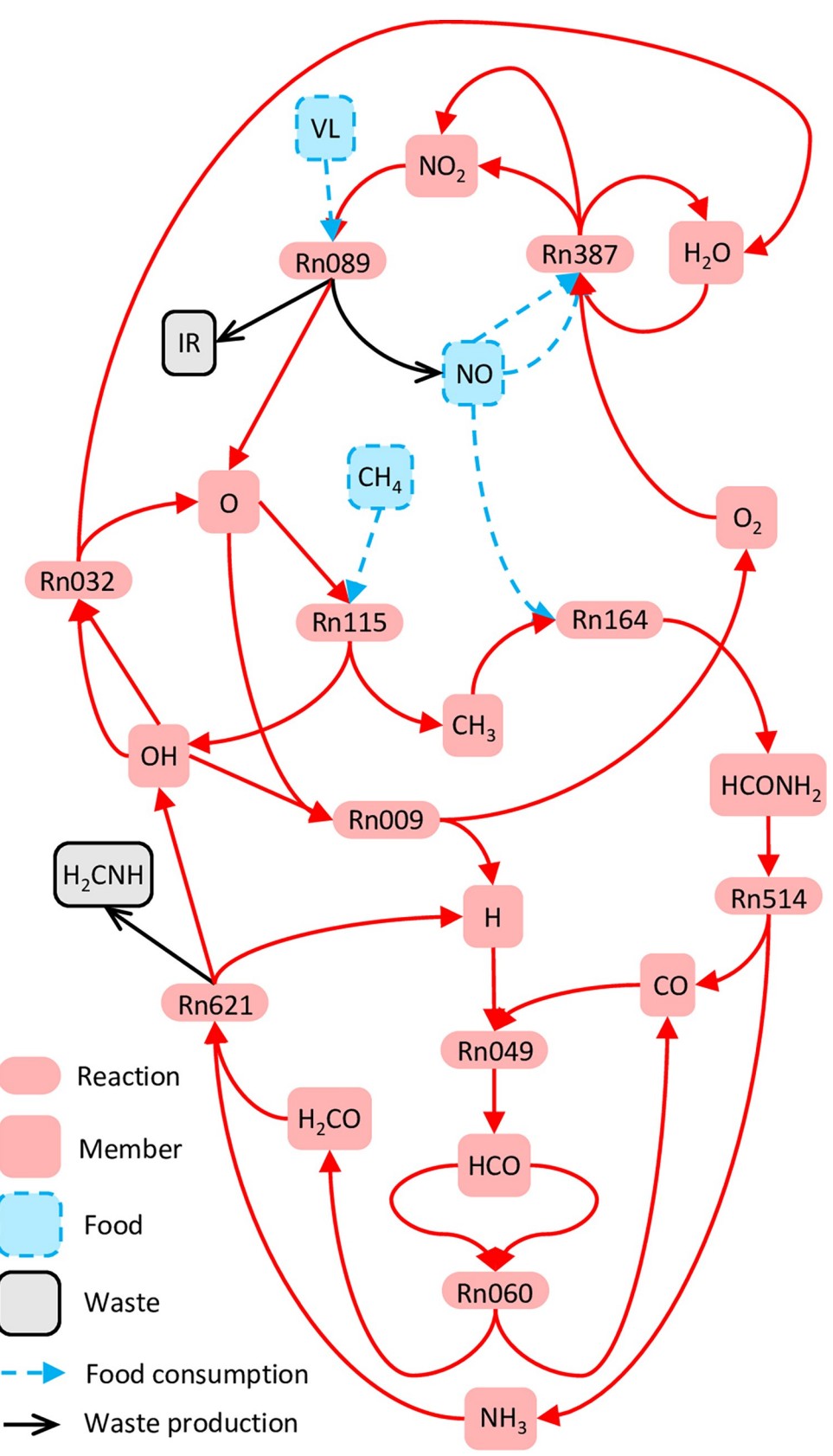

**Fig 3. A minimal autocatalytic motif within abiotic tier-1 SDAS.** This autocatalytic motif is one of the motifs detected by applying the integer programming procedure to the abiotic tier-1 SDAS, with the criterion that the motif should contain as few reaction types as possible. VL: visible light. IR: infrared light.

smallest species is the 4-carbon-atom cytosine ($C_4H_5N_3O$), can induce a 56-species, 180-reaction tier-2 SDAS (SDAS-bio-2a). SDAS-bio-2a includes all the remaining reactions in SDAS-bio-1b (S18 and S19 Tables, S4B Fig), meaning that SDAS-bio-1b could be seeded by the 2-carbon-atom glycolaldehyde followed by the 4-carbon-atom cytosine instead of by a 9-carbon-atom pyrimidine nucleoside. Likewise, we found that the composite seed {adenine($C_5H_5N_5$), picolinic acid($C_6H_5NO_2$)} can induce a 1113-species, 3378-reaction tier-2 SDAS (SDAS-bio-2b) that adds all the remaining reactions in SDAS-bio-1c (S20 Table, S4C and S4E Fig), which would have required the 21-carbon-atom nicotinamide dinucleotide to be seeded directly.

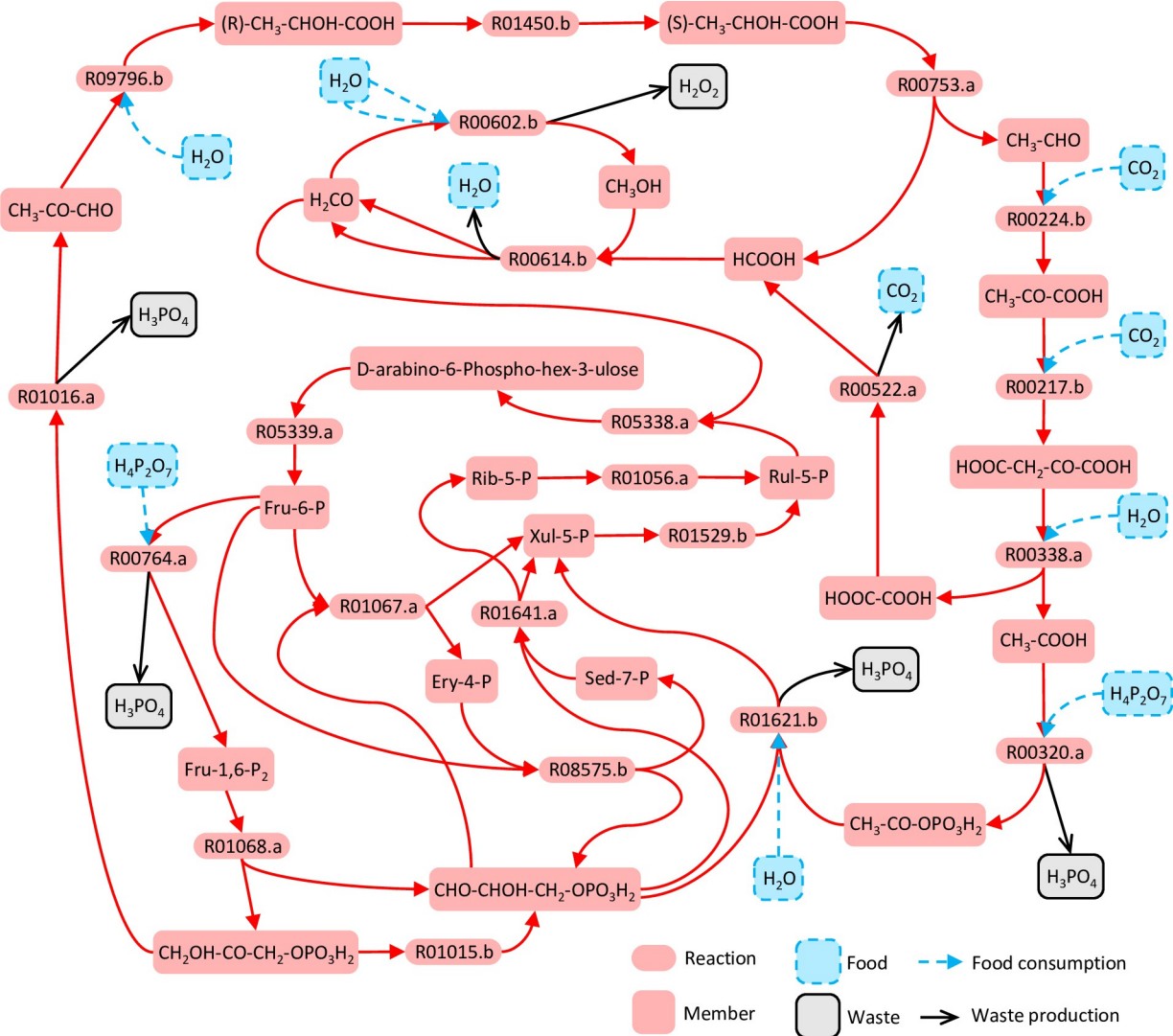

**Fig 4. A minimal autocatalytic motif within biochemical tier-1 SDASs.** This autocatalytic motif is one of the motifs detected by applying the integer programming procedure to SDAS-bio-1a, with the criterion that the motif should contain as few reaction types as possible. This motif is centered around phosphorylated monosaccharides and small carboxylic acids.

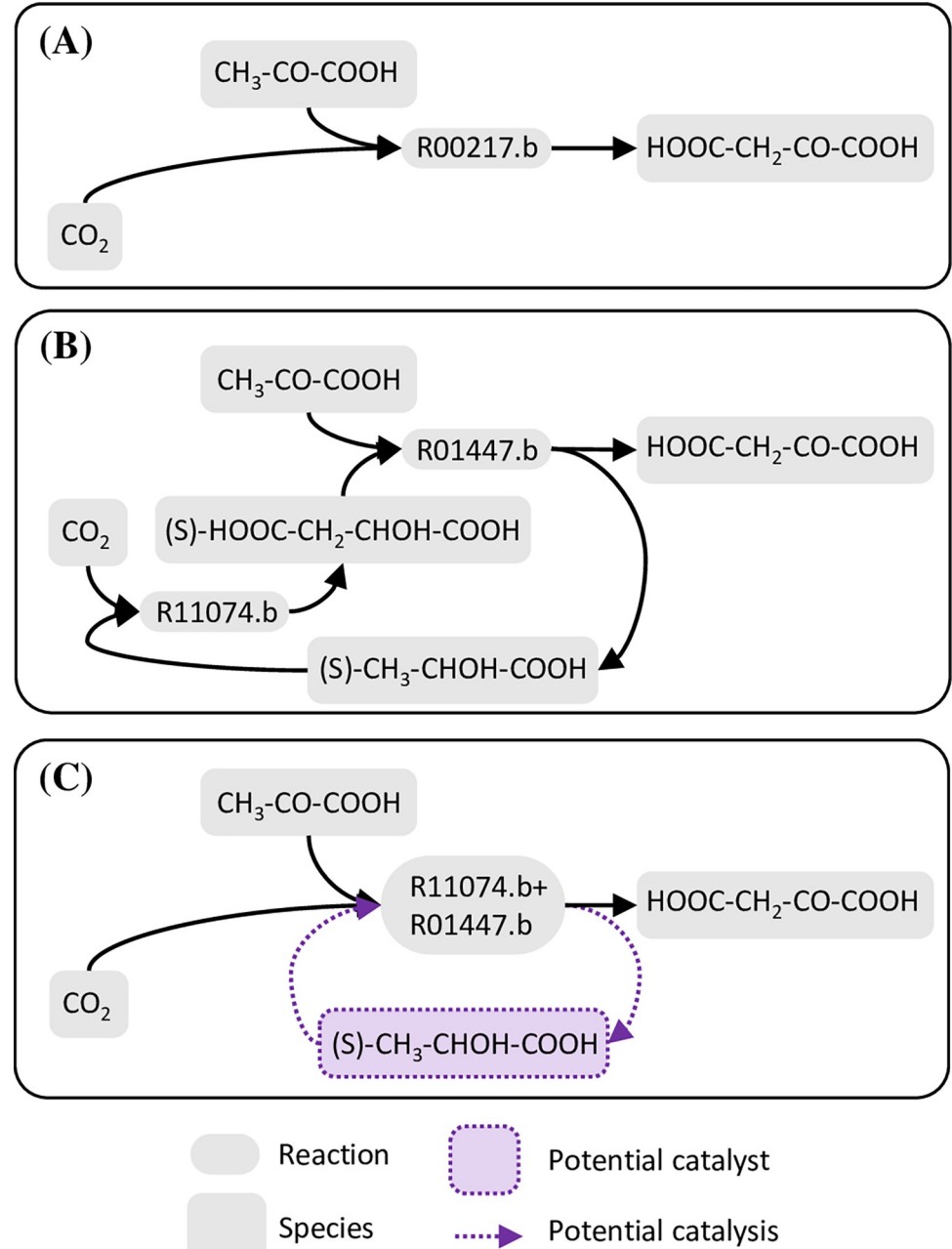

**Fig 5. (S)-lactic acid may catalyze pyruvic acid carboxylation. (A)** Direct carboxylation of pyruvic acid generates oxaloacetic acid. **(B)** Indirect carboxylation of pyruvic acid is also possible. Carbon dioxide reacts with (S)-lactic acid first to generate (S)-malic acid, and then (S)-malic acid reacts with pyruvic acid to generate oxaloacetic acid and regenerate (S)-lactic acid. **(C)** If the composite reaction rate of the direct carboxylation is lower than that of the indirect carboxylation, we may say that (S)-lactic acid can catalyze the carboxylation of pyruvic acid.

The emergence of SDAS-bio-2b following SDAS-bio-1a provides several examples of possible ecosystem-level effects. SDAS-bio-2b produces long-chain amphiphiles such as hexadecanoic acid (S20 Table), which might allow for membrane formation and a reduced rate of loss of SDAS members from local microenvironments. Likewise, SDAS-bio-2b contains an autocatalytic motif feeding on tier-0 and tier-1 species to produce ATP (S5 Fig) that provides the

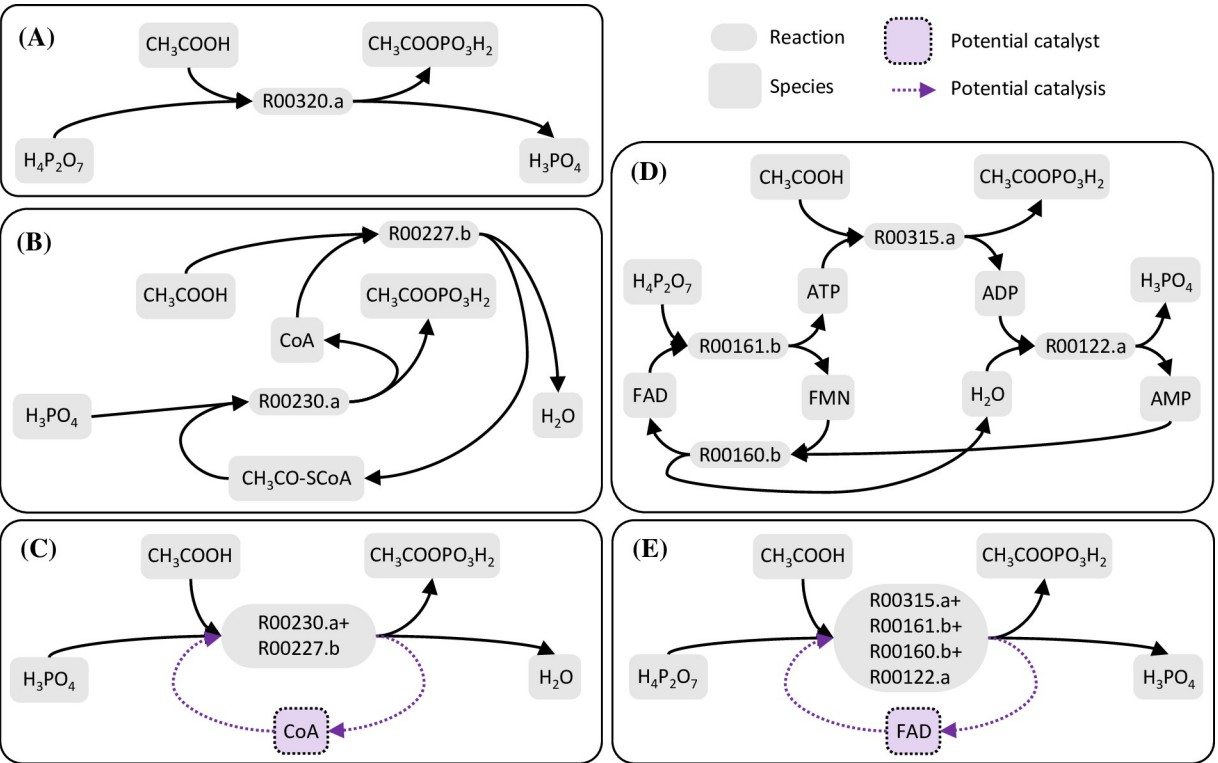

**Fig 6. Adenine-based cofactors provide alternative paths for acetic acid phosphorylation. (A)** Acetic acid can be directly phosphorylated by reacting with pyrophosphate. **(B)** With CoA present, an alternative path to phosphorylate acetic acid by consuming phosphate rather than pyrophosphate becomes possible. **(C)** We may say that CoA opens up and catalyzes a phosphate-dependent path of acetic acid phosphorylation. **(D)** With FAD present, an alternative path to phosphorylate acetic acid still by consuming pyrophosphate is possible. **(E)** If the composite reaction rate of the FAD-dependent path is faster than that of the direct phosphorylation, we may say that FAD can catalyze the pyrophosphate-dependent phosphorylation of acetic acid.

adenine moiety for various adenine-based cofactors, such as CoA, FAD, and $NAD^+$. These cofactors add several additional properties. First, they may act as potential catalysts for key reactions of SDAS-bio-1a. For example, for acetate phosphorylation (Figs 4 and 6A), CoA enables an alternative path with $H_3PO_4$ as phosphorus donor (Fig 6B and 6C), whereas FAD enables one that still uses the likely less abundant $H_4P_2O_7$ (Fig 6D and 6E). Second, higher-tier cofactors allow new network motifs to use unexploited food or use exploited food in new ways. For example, the reactions, $NAD^+ + H_2S \rightleftharpoons NADH + H^+ + S$ and $NADP^+ + H_2S \rightleftharpoons NADPH + H^+ + S$ use $H_2S$ as a hydrogen donor, which enables a new carbon-fixing autocatalytic cycle (Fig 7) that is much shorter than the SDAS-bio-1a autocatalytic motif (Fig 4, S15 Table). Third, due to the previous two properties, a possible example of scaffolding may be seen, in which the activation of a higher-tier SDAS makes the system resilient to the deactivation of lower-tier reactions that were essential in an earlier stage. For example, SDAS-bio-1a requires the activity of R00602.b or R00614.b (Fig 4), but these reactions cease being essential for auto-catalysis after SDAS-bio-2b has been activated (S21 Table).

The hierarchical organization of SDASs in the biochemical network suggests that multiple life-like properties, including catalysis by internally synthesized molecules, provision of energy-expensive monomers, compartmentalization mediated by amphiphiles, innovative ways to exploit environmental resources, and interdependence between complex modules, can be natural derivatives of sequential activation of SDASs by new seeds that are not much more complex than the set of species that already exist.

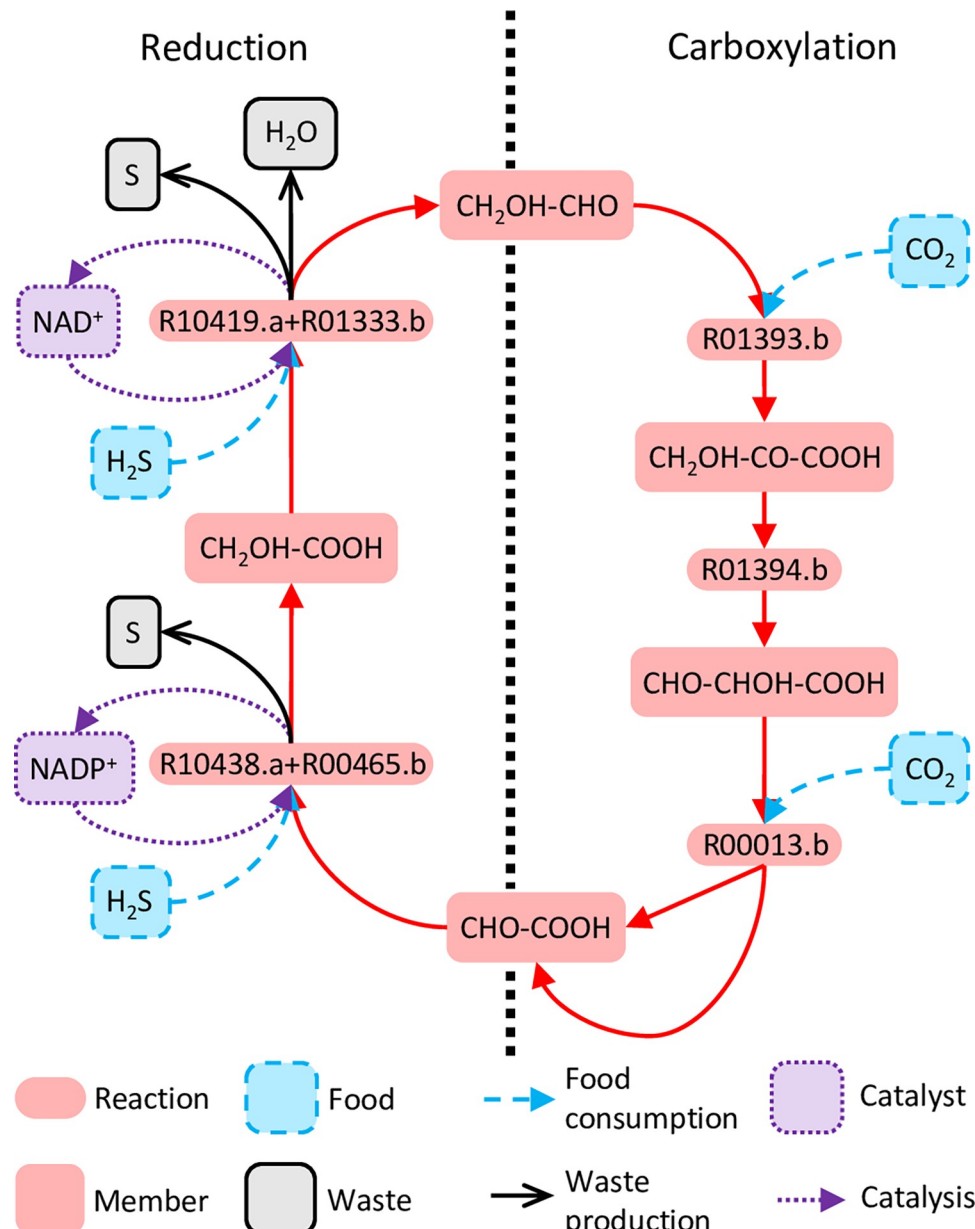

**Fig 7. A carbon-fixing autocatalytic cycle enabled by NAD(P)$^+$.** The presence of NAD(P)$^+$ enables reactions that turn $H_2S$ into possible terminal hydrogen donors for reducing glyoxylic acid to glycolic acid and finally to glycolaldehyde, which creates a stoichiometrically autocatalytic cycle that performs carbon fixation.

## 3. Discussion

### 3.1. Conclusions

We formalized the concept of SDASs, developed methods to detect and analyze them, and used the methods to explore the potential for self-maintenance and complexification in two real chemical reaction networks. We found that the abiotic and biochemical networks share multiple properties, including hierarchically organized SDASs converting simple materials and energy to more complex species. They also both included cases of unsupported seeds, composite seeds, and the potential for higher-tier SDASs to have emergent effects. The

hierarchical structure of SDASs allows seeds to exploit the complexity of previously established SDASs so that complexification can be mediated by sequential seeding of relatively simple species. Further, we noted that scaffolding may remove previously necessary network components.

## 3.2. Relationship between seeding and similar concepts

For anyone who knows Louis Pasteur's gooseneck flask experiment and accepts that a bacterium is an autocatalyst, the idea of seeding should be straightforward – an autocatalytic system can, once seeded, propagate itself in the presence of food but cannot spontaneously emerge from food alone. Multiple researchers have stressed this feature of autocatalytic systems, although with different terminology and framing, for example "obligate autocatalyst" [59], "exclusive autocatalysis" [60], and "reactions that are part of autocatalytic cycles" but are not accessible by "direct synthesis reactions" [25].

Nevertheless, it is noteworthy that our concept of an unsupported seed, a set of chemical species that seeds an autocatalytic system but is not produced by that system, has largely been overlooked by previous literature. This concept, together with the concept of scaffolding, implies that during abiogenesis, some key events possibly left no historical record in extant metabolism and thus became missing links. Additionally, although the possibility of composite seeds was considered by previous literature [59], we believe that it has not received as much attention as it warrants, given that molecular interdependence is a prominent yet puzzling feature of metabolism [61]. Stepwise complexification of reaction networks triggered by sequential introduction of seeds was also not extensively discussed in prior work. We have showed that such complexification is possible and, moreover, can be achieved with seeds that are not much more complex than the chemicals already produced by previously activated reactions. Thus, our concept of seeding provides a unifying framework that covers similar concepts described in previous literature and introduces key concepts such as unsupported seeds and the potential seeding of trophic tiers.

As we have pointed out previously [26], the seeding of autocatalytic motifs provides a basis for heritable change and, thus, evolution. When a new, viable SDAS is activated by a seed or an existing SDAS is deactivated due to environmental changes (as in the case of scaffolding), a new heritable state may arise. This makes activation and deactivation of SDASs the pre-genetic analog of mutations. Indeed, insofar as a genetic mutation entails a rare chemical reaction creating or deleting a nucleic acid sequence that is capable of autocatalytic maintenance due to the replication machinery, a genetic mutation is a special case of a chemical seed. This suggests that the concept of seed-dependent autocatalysis provides a conceptual bridge between the evolution-like dynamics of food-driven, small-molecule reaction systems and the more familiar Darwinian evolution of genetic-based systems.

## 3.3. Relationship between RAF theory, COT, and SDAS theory

At first glance, our SDAS theory may seem similar to RAF theory [23], but there are significant differences. SDAS and RAF theories define catalysis and autocatalysis in very different ways. SDAS theory focuses on stoichiometry, with catalysis interpreted generically as a case when a chemical species (i.e., catalyst) is both consumed and regenerated by one or a series of reactions. In SDAS theory, explicit catalysis is a special case where the catalyst is a reactant and product of a single reaction (e.g., Fig 3, $H_2O$ catalyzes Rn387). Under SDAS theory, an explicitly catalyzed reaction has the potential to be a "branching reaction" [26] or "fork" [17] of an autocatalytic motif because the catalyst is immediately regenerated by the reaction and other product(s) of the reaction may go on to be converted into a new copy of the catalyst. In

contrast, RAF theory in its basic form requires that every reaction in a RAF or pseudo-RAF [27] is explicitly catalyzed. As a result, a SDAS is not necessarily a pseudo-RAF. For example, with $\{F_1, F_2\}$ as food, $\{A + F_1 \rightarrow B + C, B + F_2 \rightarrow A + D, C + D \rightarrow A\}$ forms a SDAS but not a pseudo-RAF because no reaction is explicitly catalyzed. Also, since RAF theory lacks an explicit treatment of stoichiometry, a RAF or pseudo-RAF is not guaranteed to be stoichiometrically autocatalytic. For example, with$\{F_1, F_2, F_3\}$ as food, $\{F_1 \xrightarrow{F_2} A, F_3 \xrightarrow{A} B\}$ is a RAF and $\{F_1 \xrightarrow{F_2} A, B \xrightarrow{A} C, C \xrightarrow{A} B\}$ is a pseudo-RAF but neither is stoichiometrically autocatalytic.

Likewise, COT [19] and SDAS theory focus on different aspects of complex dynamical systems, although they both require stoichiometric information. COT cares about the movement of a dynamical system in state space via the appearance and extinction of components, no matter whether the environment is open or not. As a result, the self-maintenance of a dynamical system under COT does not necessarily depend on autocatalysis – a closed system or even an empty system is considered self-maintaining, and it is the movement itself, not the probability of triggering the movement or whether the movement leads to accretion of complexity and life-like properties, that plays a critical role. In contrast, SDAS theory assumes that every internal species is always directly or indirectly subject to dilution due to environmental openness such that autocatalysis is the only possible way for an internal system to self-maintain. Moreover, SDAS theory specifically cares about the chemical origin of life, so the accretion of complexity and life-like properties by events that are not too improbable needs to be mapped to the topological features of reaction networks.

### 3.4. Future directions

The similarities between the abiotic and biochemical networks allow a role for SDASs in the transition from abiotic to biotic chemistry, except that the presence of just two tiers in each network seems insufficient to account for stepwise complexification. The limited number of tiers might simply reflect the databases being tiny compared to the immensity of chemistry. Analyses of carbonaceous chondrites [62], prebiotic synthesis systems [63–65], and *in silico* chemistry [66] suggest that prebiotically relevant reaction network includes a vast number of organic compounds. In addition, for the biochemical network, old reactions might have been removed by scaffolding and new reactions might have been enabled by the evolution of enzymes, which could partially mask the ancient hierarchical structure and result in a limited number of tiers. Moreover, it is possible that stepwise complexification depends not only on network topology but also on thermodynamic and kinetic factors, which we did not consider. To evaluate these alternatives, future work will require larger real or rule-based reaction databases [66–68] and simulations that can accommodate the stochasticity that will arise when a set of highly diverse chemicals is present, with some chemicals at miniscule concentrations.

While the two networks showed many similarities, there was one noteworthy difference: the abiotic network lacked examples of higher-tier species potentially catalyzing the assimilation of the ultimate food. Since biochemical reactions depend heavily on enzymes, it is easy to imagine that reactions leading to the production of cofactors were retained by natural selection, while other potential reactions were effectively repressed due to their unfavorable kinetics (compared to the enzyme-dependent ones). Exploring this possibility would require an expanded model that includes reactions that generate potentially catalytic polymers. However, this could face considerable computational challenges due to the combinatorial explosion of species generated by polymerization cascades.

The theory of stepwise complexification by rare seeding events is eminently testable in the laboratory. The approach would be to set up open systems experiencing an input of simple food species and ongoing dilution. This would select, effectively, for autocatalytic processes

that emerge [69–71]. Then one could readily see if transient addition of complex molecules triggers the maintained production of additional chemical complexity. Finding seeded systems that remain significantly different from unseeded controls would both support the idea of seed-dependent complexification and provide a tangible example of chemical memory, a kind of heritability that could support adaptive evolution by natural selection prior to the origin of genetic encoding.

# 4. Materials and methods

## 4.1. Preprocessing databases of reactions

**4.1.1. Abiotic reaction database.** The reaction network assembled by Adam et al. includes the following categories: free radical reactions, mineral geochemical reactions, amino acid production, chloride radical and polar reactions, nitrile radical and polar reactions, RNA nucleotide assembly, nuclear decay, and physicochemical reactions [35]. We processed this database by the following steps.

First, we excluded the nuclear decay reactions because we did not plan to put radioactive atoms into the ultimate food set.

Second, with kind help from Dr. Zachary R. Adam and Dr. Albert C. Fahrenbach, we deleted duplicate reactions, added a few new reactions that were not in the original database (S1 Table), balanced some reaction equations, and excluded the reactions without clear stoichiometry. This is because our method requires stoichiometry of reactions.

Third, we added the formose reaction into the database. According to Breslow's mechanism [49], the formose reaction is driven by aldol and retro-aldol reactions and aldose-ketose isomerization. In combination these reactions allow low-carbon-number monosaccharides to generate high-carbon-number monosaccharides. Therefore, we added reversible aldol reactions and reversible aldose-ketose isomerization among formaldehyde, glycolaldehyde, and monosaccharides with no more than 8 carbon atoms. Optical isomers were not distinguished from each other. Formaldehyde dimerization was not added because it is very slow and its unclear reaction mechanism is neither aldol/retro-aldol reaction nor aldose-ketose isomerization.

Fourth, every reaction labeled reversible was split into two unidirectional reactions.

**4.1.2. Biochemical reaction database.** We processed the reaction database curated by Xavier et al. [36] to obtain the biochemical reaction database by the following steps.

First, we removed all reactions involving chemical species that do not have specific molecular mass, such as reduced ferredoxin (KEGG: C00138), acyl-carrier protein (KEGG: C00229), starch (KEGG: C00369), and long-chain aldehyde (KEGG: C00609), because they sometimes result in "fake" stoichiometric relationships. For example, the reaction: starch + $H_2O \rightleftharpoons$ dextrin + starch (KEGG: R02108) would make starch an infinite source of dextrin as long as $H_2O$ is provided. Considering that glycans have both "G"-started (meaning "Glycan") and "C"-started (meaning "Compound") KEGG entries, the reactions involving "G"-started entries were also removed because these reactions are redundant.

Second, we added some obviously spontaneous reactions that were missing, such as $H_2O \rightleftharpoons H^+ + OH^-$ and $H_2CO_3 \rightleftharpoons H^+ + HCO_3^-$.

Third, we added the reaction KEGG R06974 into the biochemical reaction database. This reaction is very similar to the reaction R06975 (S2 Fig): both reactions use HCOOH as the carbon donor to add a -CHO to -$NH_2$ and form an -NH-CHO with ATP hydrolysis providing energy for the reaction. However, R06975 is in the network curated by Xavier et al. while R06974 is not [36], presumably because the annotations of R06974 in the KEGG database are not as detailed as those of R06975, and thus R06974 was filtered out. We also conducted

analyses without R06974. In that case, SDAS-bio-1a, SDAS-bio-1b, and SDAS-bio-2a were not affected but SDAS-bio-1c and SDAS-bio-2b were missing. However, we opted to present results that included the plausible reaction R06974 so as to better illustrate the potential for ecosystem-level feedback without resorting to analyzing the entirety of KEGG.

Fourth, as all reactions in the KEGG biochemical reaction database are labeled reversible, every reaction was split into two unidirectional reactions. The reaction following the forward direction specified in the KEGG database has a suffix ".a" to its entry, and that of the reverse direction has a suffix ".b".

Fifth, the reactions that are labeled as multi-step were removed because each step is already a reaction in the database. Although keeping these multi-step reactions may not have big impact on the detection of SDAS existence, decreasing the number of reactions in the stoichiometric matrix should help accelerate the computation.

## 4.2. Network expansion

The set $R = \{r_1, r_2, \ldots, r_i, \ldots, r_I\}$ is a set of multiple reactions $r_i$'s that are allowed. Each $r_i$ specifies reactants and products, and the union of all reactants and products across all $r_i$'s is the maximum set of chemical species $S = \{k_1, k_2, \ldots, k_j, \ldots, k_J\}$. We define an operation called network expansion, $\Xi(S_O, R) = (S_E, R_E)$, where $S_O$ is the subset of $S$ from which the expansion starts, $S_E$ the set of chemical species resulting from the expansion, and $R_E$ the set of reactions resulting from the expansion. The expansion is conducted as follows:

i.  Let $R_E = \varnothing$; define a set of reactions $R' = R$; let $S_E = S_O$.

ii. Define a temporary set of chemical species $S' = \varnothing$.

iii. For a reaction $r_i$ in $R'$, check if the reactants (and catalysts, if applicable) required by $r_i$ are all present in $S_E$; if so, move $r_i$ from $R'$ to $R_E$, and scan through the products of $r_i$ to add the chemical species that are not in $S_E$ to $S'$. Do this for all reactions in $R'$. Then add all chemical species in $S'$ to $S_E$. If during this step, no reaction in $R'$ is moved, then the expansion is finished; otherwise, proceed to (ii).

## 4.3. Identifying cliques

Let us assume that $S_F$ is a set of chemical species resulting from a network expansion within the set of allowed reactions $R$ from a set of ultimate food $S_{UF}$. Two non-empty supported seeds $H_1$ and $H_2$ are said to be in the same clique if (a) $\Xi(S_F \cup H_1, R) = \Xi(S_F \cup H_2, R)$, and (b) $\Xi(S_F \cup H_1', R) \neq \Xi(S_F \cup H_1, R), \forall H_1' \subset H_1$ and $\Xi(S_F \cup H_2', R) \neq \Xi(S_F \cup H_2, R), \forall H_2' \subset H_2$.

In this paper, we only investigated the cliques consisting of supported seeds that are individual chemical species (i.e., singleton supported seeds) for two reasons: first, unsupported seeds may induce the same SDAS but different non-SDAS reactions and species; second, exhaustively enumerating all composite seeds could be computationally prohibitive. Nonetheless, the principle could be expanded to potential supported seeds comprising more than one chemical species.

## 4.4. Detecting seed-dependent autocatalytic systems (SDASs) by linear programming

Let us assume that a $(p-1) \times (q-1)$ stoichiometric matrix, where each row represents a chemical species and each column represents a unidirectional reaction, results from a network expansion within the set of allowed reactions $R$. The row labels of this $(p-1) \times (q-1)$ stoichiometric matrix form a chemical species set $S_F = \{k_1, k_2, \ldots, k_{p-1}\}$, which is defined as the

external food for this detection process. Now we select a non-empty set of non-food chemical species $H = \{k_p, k_{p+1}, \ldots, k_{p+h}\}$ to serve as the candidate supported seed. For example, we can treat all chemicals not in the external food as candidate singleton supported seeds and search through these one at a time. Then we conduct a network expansion from the food set and the candidate supported seed, $\Xi(S_F \cup H, R) = (S_{FH}, R_{FH})$, generating $S_{FH} = \{k_1, k_2, \ldots, k_m\}$ and $R_{FH} = \{r_1, r_2, \ldots, r_n\}$.

A SDAS feeding on $S_F$ exists if there is a vector of non-negative elements $x = (x_q, x_{q+1}, \ldots, x_n)$ such that Eq (1) from Sect. 2.2. is fulfilled. To determine whether such an $x$ exists, we used the linear programming tool provided by SciPy v1.6.2 (https://docs.scipy.org/doc/scipy/reference/generated/scipy.optimize.linprog.html). In addition to the constraint set by (1), this linear programming tool requires an objective function. Since we knew that the growth of an autocatalytic system feeding on the external food is unbounded when the external food is unlimited, we could set an objective function to find the maximum $\sum_{j=q}^{n} x_j s_{ij}$ for an internal species $k_i$ ($i \in [p, m]$). If this objective function was found to be unbounded, we knew that a feasible region constrained by (1) must exist, indicating that a SDAS must exist. We simply let $i = p$ and set

$$\max_{x_q, \ldots, x_n} \sum_{j=q}^{n} x_j s_{pj} \tag{2}$$

as the objective function.

We used the "highs" method [48] to confirm the existence of SDASs. Once the SDAS was confirmed to exist, we ran the integer programming process, described in Sect. 4.5, to find autocatalytic motifs within the SDAS, subject to further specific constraints.

## 4.5. Detecting minimum-reaction autocatalytic motifs by integer programming

We can use linear programming to identify a SDAS, but we also desire to find small autocatalytic motifs within each SDAS, because these are easier to visualize and could potentially guide future experimental studies. Specifically, considering that there were likely few types of catalysts in the prebiotic world, we focus on finding the autocatalytic motifs with as few reaction types as possible. As a result, we want to minimize the number of positive components of $x$ while the reactions corresponding to positive $x_j$'s still form an autocatalytic motif feeding on the external food. In fact, the original SDAS may contain multiple autocatalytic motifs. In this section we describe a method based on integer linear programming to enumerate small-cardinality autocatalytic motifs within a given SDAS.

If there exists a vector $x$, such that (1) is fulfilled, then by scaling $x$, we may ensure that

$$\sum_{j=q}^{n} x_j s_{ij} \geq 1 \ (x_j \geq 0) \ \forall i \in [p, m]. \tag{3}$$

Thus, without loss of generality, we may use (3) as the constraint. There may be many possible $x$'s that fulfill (3), and we used integer programming to seek and enumerate SDASs with desirable properties.

To find smaller systems, we seek a set of columns $T \subseteq [q, n]$ such that if $j \in T$ and $\exists i \in [p, m]$ which makes $s_{ij} \neq 0$, then $\exists x_j \in \mathbb{R}_+^{n-q+1}$ such that $\sum_{j=q}^{n} x_j s_{ij} \geq 1$. Of course, the set $T$ should have $|T| \geq 1$. Finding such a set $T$ can be accomplished in a systematic manner by seeking solutions to a linear-inequality system wherein some of the variables are required to take integer values.

In the formulation, we use binary variables $z_j \in \{0, 1\}$ that take the value 1 if and only if column $j \in [q, n]$ is in the set $T$, and we will minimize $\sum_{j=q}^{n} z_j$ to minimize the cardinality of the

selected set. Because an autocatalytic motif must have at least one reaction, it is obvious that $\sum_{j=q}^{n} z_j \geq 1$.

Let $\beta_j$ be the upper bound on the number of the reaction $r_j$ that can occur in an autocatalytic motif. We must enforce that if column $j$ is not selected for the autocatalytic motif (i.e., $z_j = 0$), then its level of reaction $x_j$ must also be zero, which is done with the algebraic constraints $x_j \leq \beta_j z_j$.

For the selected set of reactions $T$ to be an autocatalytic motif, let us first define a set $\Omega_T$ that contains and only contains the row indices of all chemical species that are involved in the reactions in $T$ and are not external species (i.e., $\Omega_T \subseteq [p, m]$), then we must make sure that

$$\sum_{j=q}^{n} x_j s_{ij} \geq 1 \ \forall i \in \boldsymbol{\Omega}_T. \tag{4}$$

This is done by introducing additional binary variables $y_i \in \{0, 1\}$ ($i \in [p, m]$) indicating if species $k_i$ is involved in the autocatalytic motif represented by the positive components of the vector $\boldsymbol{z} = (z_q, z_{q+1}, \ldots, z_n)$. Then, the following set of linear inequalities accomplish the conditions in (4)

$$\sum_{j=q}^{n} x_j s_{ij} \geq 1 - M_i(1 - y_i) \ \forall i \in [p, m], \tag{5}$$

where

$$M_i = 1 - \sum_{j=q:s_{ij}<0}^{n} \beta_j s_{ij}. \tag{6}$$

To understand how (5) works, imagine that $k_i$ is involved in the autocatalytic motif, then $y_i = 1$ and (5) is equivalent to (4). In contrast, if $k_i$ is not involved in the autocatalytic motif, then $y_i = 0$ and (5) is equivalent to

$$\sum_{j=q}^{n} x_j s_{ij} \geq \sum_{j=q:s_{ij}<0}^{n} \beta_j s_{ij}. \tag{7}$$

Because $\beta_j$ is the upper bound on $x_j$, (7) should always hold and thus it is redundant in the linear system, representing the fact that (4) does not need to be considered for a $k_i$ that is not included in the autocatalytic motif.

It is necessary to link a reaction and the chemical species that are involved in the reaction. If a reaction $r_j$ ($j \in [q, n]$) is selected for an autocatalytic motif (i.e., $z_j = 1$), any internal chemical species $k_i$ that is involved in $r_j$ must also exist in the autocatalytic motif (i.e., $y_i = 1$). Therefore, for any reaction $r_j$ ($j \in [q, n]$), we define a set $\Omega_j$ that contains and only contains the row indices of all chemical species that are involved in $r_j$ and are not external species (i.e., $\Omega_j \subseteq [p, m]$). Then we apply the constraint

$$y_i \geq z_j \ \forall j \in [q, n], \forall i \in \boldsymbol{\Omega}_j \tag{8}$$

which guarantees that once $r_j$ is included in an autocatalytic motif (i.e., $z_j = 1$), $k_i$ needs to be sustainably synthesized (i.e., $y_i = 1$).

This gives a full integer programming formulation for finding a minimum-cardinality autocatalytic motif among the reactions $\{r_q, r_{q+1}, \ldots, r_n\}$. We set the integer programming problem

as to find

$$\min_{\boldsymbol{x,y,z}} \sum_{j=q}^{n} z_j \qquad (9)$$

which is constrained by

$$
\begin{cases}
x_j \geq 0 \; \forall j \in [q, n] \\
z_j \in \{0, 1\} \; \forall j \in [q, n] \\
y_i \in \{0, 1\} \; \forall i \in [p, m] \\
\displaystyle\sum_{j=q}^{n} z_j \geq 1 \\
x_j \leq \beta_j z_j \; \forall j \in [q, n] \\
\displaystyle\sum_{j=q}^{n} x_j s_{ij} \geq 1 - M_i(1 - y_i) \; \forall i \in [p, m] \\
y_i \geq z_j \; \forall j \in [q, n], \forall i \in \boldsymbol{\Omega}_j
\end{cases}
\qquad . \qquad (10)
$$

This formulation can be solved computationally using a state-of-the-art integer programming software, such as Gurobi [72]. The positive elements of the binary solution vector $\boldsymbol{z}$ indicate the reaction set $\boldsymbol{T}$ in a minimum-cardinality autocatalytic motif chosen from the set of reactions $\{r_q, r_{q+1}, \ldots, r_n\}$.

It should be noted that multiple autocatalytic motifs with the same number of reaction types may exist, and our integer programming can enumerate all minimum-cardinality autocatalytic motifs by solving a sequence of integer programs. After a binary solution vector $\boldsymbol{z}$ and its associated reaction set $\boldsymbol{T}$ are identified, the constraint $\sum_{j=q:j \in T}^{n} z_j \leq |\boldsymbol{T}| - 1$ may be added to (10), and then the process repeats. Furthermore, if we want to find minimum-cardinality autocatalytic motifs with at least $D$ reactions, we can do it by replacing $\sum_{j=q}^{n} z_j \geq 1$ with $\sum_{j=q}^{n} z_j \geq D$ in (10).

## Supporting information

**S1 Fig. Network expansion and seeding.** The operation of network expansion and seeding can be illustrated by an expanding stoichiometric matrix, where each row represents a chemical species and each column represents a reaction, with the stoichiometric coefficients as entries. **(A)** The expansion starts from a set $S_U$ of "ultimate food" species, which are assumed to be provided by the environment on an ongoing basis. **(B)** The reactions where the reactants are all provided by the existing rows are added as new columns. **(C)** If these newly added reactions introduce some new chemical species other than the ones represented by the existing rows, these new chemical species are added as new rows. **(D)** Such iterative addition of columns and rows continues until no more columns can be added, completing an expansion, which generates the tier-0 system $(S_0, R_0)$. **(E)** A new set of chemical species $H$ is added as the candidate supported seed. **(F)** Reactions where the reactants are all provided by the candidate supported seed and the tier-0 system are added as new columns. **(G)** These newly added reactions introduce some new chemical species other than the ones represented by the existing rows, and these new chemical species are added as new rows. **(H)** Iterative addition of columns and rows continues until no more columns can be added, completing an expansion, and a tier-1 system $(S_1, R_1)$ is defined as the complement of the tier-0 system in the results of this

expansion triggered by the candidate supported seed.
(TIF)

**S2 Fig. The KEGG reactions R06974 and R06975 are similar.** These two reactions are highly similar in terms of how -NH$_2$ is modified to -NH-CHO. The reaction schemes are downloaded from the KEGG reaction database and modified by adding blue contours to emphasize the relevant moieties.
(TIF)

**S3 Fig. Distributions of the numbers of reactions that a chemical species is involved.** Each species can be involved in one or multiple reactions. **(A)(C)** The number of reactions involving the focal species is counted for every species, and then this statistic is plotted as a histogram. **(B)(D)** Then the data of a histogram is plotted in a new graph where the x-axis shows the natural logarithm of the central value of the binned reaction number and the y-axis shows the natural logarithm of the number of chemical species. The results of simple linear regression and correlation coefficients are also shown. Note that the last bin of a histogram is not shown in the corresponding logarithmic graph because it actually represents all bins with larger numbers of reactions rather than a single bin, and that no data point is shown for the bins with zero count in the logarithmic graph because the logarithm of zero is undefined. **(A)(B)** Abiotic reaction database. **(C)(D)** Biochemical reaction database.
(TIF)

**S4 Fig. Sequential seeding of SDASs in the biochemical network.**
(TIF)

**S5 Fig. The key autocatalytic cycle within the autocatalytic motif synthesizing ATP within SDAS-bio-2b.** This autocatalytic cycle requires prior establishment of a lower-tier system (SDAS-bio-1a) able to supply chemicals such as glycine and formic acid. Note that reaction R00156.b is used repeatedly in this cycle, and that some chemicals, such as 5-phosphoribosylamine and UTP, are synthesized from tier-0 and SDAS-bio-1a chemicals by SDAS-bio-2b reactions that are not shown in this graph. Note that some waste of the entire autocatalytic cycle (e.g., H$_2$O) can be the food for a reaction step (e.g., R04463.a).
(PPTX)

**S1 Table. Abiotic reaction database.**
(XLSX)

**S2 Table. Biochemical reaction database.**
(XLSX)

**S3 Table. Scanning the abiotic reaction network for tier-1 singleton supported seeds.**
(XLSX)

**S4 Table. The tier-1 SDAS found in the abiotic reaction network.**
(XLSX)

**S5 Table. Examples of tier-1 autocatalytic motifs found in the abiotic reaction database.**
(XLSX)

**S6 Table. HCO and O$_2$ form a composite supported seed in the abiotic reaction network.**
(XLSX)

**S7 Table. Singleton unsupported seeds for the abiotic tier-1 SDAS.**
(XLSX)

**S8 Table. Scanning the abiotic reaction network for tier-2 singleton supported seeds.**
(XLSX)

**S9 Table. The tier-2 SDAS found in the abiotic reaction network.**
(XLSX)

**S10 Table. The biochemical tier-0 system.**
(XLSX)

**S11 Table. Scanning the biochemical reaction network for tier-1 singleton supported seeds.**
(XLSX)

**S12 Table. The tier-1 SDAS (SDAS-bio-1a) seeded by the 267-member clique, found in the biochemical reaction network.**
(XLSX)

**S13 Table. The tier-1 SDAS (SDAS-bio-1b) seeded by the 34-member clique, found in the biochemical reaction network.**
(XLSX)

**S14 Table. The tier-1 SDAS (SDAS-bio-1c) seeded by the 3-member clique, found in the biochemical reaction network.**
(XLSX)

**S15 Table. Examples of minimal autocatalytic motifs identified within the tier-1 SDAS in the biochemical reaction database.**
(XLSX)

**S16 Table. Formaldehyde and acetate form a composite supported seed in the biochemical reaction network.**
(XLSX)

**S17 Table. Singleton unsupported seeds for biochemical tier-1 SDASs.**
(XLSX)

**S18 Table. Scanning the biochemical reaction network for tier-2 singleton supported seeds.**
(XLSX)

**S19 Table. The tier-2 SDAS (SDAS-bio-2a) found in the biochemical reaction network.**
(XLSX)

**S20 Table. The tier-2 SDAS (SDAS-bio-2b) found in the biochemical reaction network, seeded by interdependent adenine and picolinic acid.**
(XLSX)

**S21 Table. NADH can induce a SDAS without reactions R00602.b and R00614.b.** Once SDAS-bio-2b is activated, its sustainably produced species NADH can induce a SDAS even with deactivated reactions R00602.b and R00614.b, which are necessary for the species in the bio-1a clique to induce SDAS-bio-1a.
(XLSX)

**S1 Python Script. Network expansion.**
(ZIP)

**S2 Python Script. Linear programming.**
(ZIP)

**S3 Python Script. Integer programming.**
(ZIP)

## Acknowledgments

We thank Zachary Adam and Albert Fahrenbach for their kind help with curating the abiotic reaction database and the following for useful discussions: Alyssa Adams, Stephanie Colón-Santos, Emily Dolson, Praful Gagrani, Juan Perez Mercader, Alex Plum, Daniel Segrè, D. Eric Smith, Lena Vincent, and participants in the Evolving Chemical Systems workshop at the Santa Fe Institute, organized by Chris Kempes and Oana Carja (Nov. 2019).

## Author Contributions

**Conceptualization:** Zhen Peng, David A. Baum.

**Data curation:** Zhen Peng.

**Formal analysis:** Zhen Peng, Jeff Linderoth.

**Funding acquisition:** David A. Baum.

**Investigation:** Zhen Peng.

**Methodology:** Zhen Peng, Jeff Linderoth.

**Software:** Zhen Peng, Jeff Linderoth.

**Supervision:** David A. Baum.

**Writing – original draft:** Zhen Peng, David A. Baum.

**Writing – review & editing:** Zhen Peng, David A. Baum.

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
