## [Decision Letter · Decision Letter 0]

11 Jul 2022

Dear Dr. Baum,

Thank you very much for submitting your manuscript "The hierarchical organization of autocatalytic reaction networks and its relevance to origin of life" for consideration at PLOS Computational Biology.

As with all papers reviewed by the journal, your manuscript was reviewed by members of the editorial board and by several independent reviewers. In light of the reviews (below this email), we would like to invite the resubmission of a significantly-revised version that takes into account the reviewers' comments.

You will find attached the comments of the two reviewers.

Please provide a point-by-point answer and correspondingly revise the text where necessary.

You should notably address the concerns of reviewer 2 by better clarifying the relationship of your work with RAF-like formalisms as well as the notion of evolvability in the context of your work.

We cannot make any decision about publication until we have seen the revised manuscript and your response to the reviewers' comments. Your revised manuscript is also likely to be sent to reviewers for further evaluation.

Sincerely,

Philippe Nghe

Guest Editor

PLOS Computational Biology

James O'Dwyer

Deputy Editor

PLOS Computational Biology

Reviewer's Responses to Questions

**Comments to the Authors:**

Reviewer #1: Peng et al SDAS

This paper discusses autocatalytic networks as they might occur at the time of the origin of life. I think the paper is interesting and quite useful. I will identify myself (Paul Higgs) as I want to discuss how this relates to my own recent work [1].

The essence of the SDAS idea in this paper seems to be that molecules which are part of the autocatalytic reactions cannot form spontaneously but they can maintain themselves if they are started by a seed. The example on line 144 is A + F -> 2A. Once A is present, it can maintain itself given a supply of F, but the spontaneous reaction F -> A does not happen. Therefore we require a seed of A in order to start the autocatalytic reaction.

This concept seems very close to several previous ideas. Andersen et al [2] use the term "exclusive autocatalysis". Quoting their paper: we "require in addition, that an autocatalytic species x cannot be produced from within in the network unless a minute amount is already present at the outset. In other words the network in question does not contain a pathway that produces x in a non-autocatalytic manner from the same food set." A very similar idea also arises in [1]. Here I distinguish direct synthesis reactions (which make autocatalytic molecules directly from food molecules) from reactions that are part of autocatalytic cycles. The Case 1 example discussed in [1] satisfies my criteria for a metabolism only because the direct synthesis reactions are excluded. In Case 3, where the direct reactions are included, the network is no longer a metabolism. Kun et al [3] use the term "obligate autcatalyst" – this is a molecule that has to be added to the food set in order to kick start the metabolism. They find that ATP is an obligate autcatalyst – "our results show that the autocatalytic synthesis of ATP is unlikely to be bypassed by other reactions in a larger network". This seems to be the same as a seed in the current paper. Lauber [5] on minimal metabolism also seems to be relevant. These ideas should be discussed in this paper, and it should be carefully described whether the concepts are the same. We are in danger of having several almost equivalent definitions.

I have made an additional point in [1] that I don't think is captured by the SDAS definition. In order to have a metabolism inside an organism (say a protocell or lipid vesicle at the time of origin of life) then the autocatalytic reactions happening inside the cell must NOT happen outside the cell. Nevertheless the inside is at least partially in contact with the outside, because we require diffusion of food molecules into the cell and export of waste molecules. The metabolism inside has to maintain a state of dynamic flow, which is not at equilibrium. This is most easily achieved when the reaction system is bistable. There must be a stable state of low (or zero) concentration of catalysts outside the cell, and a different stable state of high catalyst concentration inside the cell. In the example A + F -> 2A, we might both A and F present inside the cell, and only F outside. However, the state with only F is not a stable state, because addition of a minute amount of A will start the reaction going outside. If the autocatalytic reaction begins outside as well, then both the inside and outside will proceed to equilibrium and there will be no further metabolic turnover. The inside is only alive if the outside is dead! Ref [1] shows that Case [1] is bistable, and Case 2 is autocatalytic but only monostable. If we imagine a population of vesicles in which autocatalytic reactions are happening, there is bound to be some minimal permeability of the membrane to catalysts, or there might be an occasional bursting of a cell which releases catalysts to the exterior. If the external reaction system is not stable to such a minor perturbation, then the internal system cannot maintain itself. In the SDAS language, this is the question of whether a vanishingly small amount of the seed is sufficient to start the reaction or whether a large amount of one or more seed molecules is necessary to cause the system to jump into a new steady state.

The idea that the origin of life requires jumping from a non-autocatlytic state into an autocatalytic steady state was discussed previously in other contexts [6,7]. This can occur stochastically in a finite system. Ref [7] also discusses nested autocatalytic sets, which sounds a lot like the hierarchical arrangement in the current paper.

One thing we seem to agree on is that the RAF theory, which requires all reactions to be explicitly catalyzed, is not a good starting point for a theory of metabolism. The criticisms of RAF theory in section 3.2 seem to parallel those in [1].

Lines 62-73 – You argue that the RNA world theory is wishful thinking and that there must be a metabolism first. I don't agree with this. There is a requirement is that there must be abundant RNA monomers for an RNA world to start. But the synthesis of the monomers does not need to be autcatalytic. If there is a supply of monomers by a direct reaction pathway, then the templating reactions for RNA synthesis are inherently autocatalytic. Therefore non-enzymatic RNA replication can satisfy the requirements for an autocatalytic reaction set that maintains metabolism and growth of cells (as shown in the later part of [1]). The autocatalytic steps among small molecule metabolytes might then arise later as the result of reactions catalyzed by biopolymers (ribozymes and enzymes). Obviously RNA replication did begin at some point, whether one conceives of RNA replication as difficult or easy to begin. It does not make sense to say that the whole idea of RNA replication is wishful thinking. It does not seem to me that the prior existence of a small molecule autocatalytic set (if there were one) would make it any easier for RNA replication to begin. We know that the current pathways of nucleotide synthesis are not part of autocatalytic cycles, although they may be side reactions that connect in some way to small molecule autocatalytic cycles. It does not matter whether the nucleotide monomers are made by a direct non-autocatalytic pathway or whether they are side reactions of a small molecule cycle. RNA replication still has to start somehow. Simply saying that you think RNA replication is difficult is not a good reason for saying there was a metabolism first.

Line 318 – you have an inorganic reaction network of free radicals and geochemical reactions etc, but for some reason RNA nucleotide assembly is included in this set. This looks odd. Why mix nucleotides with inorganic reactions, if other biochemicals (sugars, amino acids etc) are not included?

In fig 3 there are some odd chemical species – O, H, OH, HCO – not O2 H+ OH- H2CO. Where are these reactions supposed to be happening? Is this inorganic reaction database a useful database for prebiotic chemistry?

[1] Higgs PG. When is a reaction network a metabolism? Criteria for simple metabolisms that support growth and division of protocells. Life 2021, 11: 966

[2] Andersen JL, Flamm C, Merkle D, Stadler PF. Defining autocatalysis in chemical reaction networks. arxiv.org 2021

[4] Kun A, Papp B, Szathmary E. Computational identification of obligatory autocatalytic repliators embedded in metabolic networks. Genome Biology 2008, 9:R51.

[5] Lauber N, Flamm C, Ruiz-Mirazo K. Minimal metabolism: a key concept to investigate the origins and nature of biological systems. BioEssays 2021; 43:2100103

[6] Wu M and Higgs PG The origin of life is a spatially localized stochastic transition. Biology Direct 2012 7:42.

[7] Giri V, Jain S. The origin of large molecules in primordial autocatalytic reaction networks. PLOS ONE 2012 7(1) e29546

Reviewer #2: The manuscript by Baum and co-workers describes an interesting idea: if there is a hierarchy of autocatalytic reaction networks then they might have been triggered by rare seeding events and thus chemical complexification of the environment (the available materials) could proceed. There is also a possibility that a lower tier autocatalytic network is eaten up by a higher one. The manuscript is very clearly and well-written.

Apart from the good points mentioned above, there are some major flaws in the method and a lot of the relevant literature is not cited.

The RAF sets are important as they have the potential to become collectively autocatalytic entities and thus exhibit heredity (see [31]). The authors just assumed that their network works without catalysts (lines 328–331). The requirement for the inclusion of the production of the catalysts in RAF is actually a strength, and a model not having this has a major flaw. Please note, that while in analysis of metabolic networks, the protein catalysts are seldom explicitly considered to be produced, there are reactions to produce peptides. Thus, one can analyse if peptides can be produces and the fluxes going in that direction.

I do not see how there is heredity in the system described by the authors. An autocatalytic cycle, given enough seed molecules, can maintain itself. One can see it as heredity, but then could any variation be passed on? Can I extend one of the constituents by a methylene-group, and then everything would go on as previously except a longer molecule is produced in the end? If not then this is a very limited form of heredity which is insufficient for evolution (you can see a more in-depth discussion of it in Zachar & Szathmáry 2010 BMC Biology 8:21 https://doi.org/10.1186/1741-7007-8-21). As for potentially evolving chemical systems see the review by Adamski et al. 2020 Nat Rev Chemistry 4:386 doi: 10.1038/s41570-020-0196-x.

The KEGG database is not really curated, if one wants to do serious work on reaction networks, then use the MetaCyc database as it has information on the directionality of reactions. Not all reactions are reversible under normal circumstances. This can cause serious problems as, for example, ATP can be produced in ways not possible in nature. Just to give you an example, reaction R00004 is irreversible, it is the hydrolysis of diphosphate to phosphate. If we allow phosphate to coalesce into diphosphate, then via R00122.b we can get ATP. So inorganic phosphate can produce ATP from AMP, which would result in a perpetuum mobile.

The network expansion operation on which the methodology relies was described first (to my knowledge) by Heinrich and co-workers (Ebenhöh et al. 2004, Genome Informatics 15:3, doi: PMID: 15712108; Handorf 2005, JME 61:498, doi:10.1007/s00239-005-0027-1). Strangely these papers are not mentioned in the manuscript. There are quite some papers out there analysing metabolic networks by this method (see for example Kun et al. 2008 Genome Biology 9:R51 doi: 10.1186/gb-2008-9-3-r51). Also autocatalytic set searching in metabolic network has been automatized by the same group as in [59]: Andersen 2020 J Sys Chem 8:121; Andersen 2012 J Sys Chem 3:1. I also missed citation to von Kiedrowski’s works on autocatalysis and how to detect them in chemical systems.

Based on the above, I suggest to reject this manuscript.

Minor comments

The abstract is too technical, and assumes that the reader is very deeply into both origin of life research and metabolic network theory. The scientific abstract should also be more accessible to the general reader.

P8 when describing clique, I do not understand the example. B is a singleton seed, as B can give rise to A+F, and B+F -> A. But A in itself cannot produce any of the other compounds. If the reaction B+F -> A is reversible (B+F <-> A) then A also becomes a seed and then I would understand why A and B form a clique.

When citing Adam et al. and Xavier et al. there is no need to have the year in parenthesis afterwards. The bracketed citation of [32] and [33] is sufficient.

P17L365 What is C2H5? It seems to be an ethyl group, but how would that yield OH (hydroxyl ion?)? S3 Table is no help here.

P17L387 a subscript formatting is missing here

Figure 4: a subscript formatting is missing in CH3-CHO (top-right part)

P27L583 The amine-group’s NH2 should be in lower index

If a subscript is an abbreviation or derivative of a word and it is not a variable then it should not be italicized. So the E in SE (line 603) or the UF in SUF (line 614). Please correct throughout the text.

P29L635 Equation 1 has been repeated here. Just looks odd.

**Have the authors made all data and (if applicable) computational code underlying the findings in their manuscript fully available?**

Reviewer #1: Yes

Reviewer #2: Yes

PLOS authors have the option to publish the peer review history of their article (what does this mean?). If published, this will include your full peer review and any attached files.

Reviewer #1: **Yes: **Paul Higgs

Reviewer #2: No
---

## [Decision Letter · Decision Letter 1]

18 Aug 2022

Dear Dr. Baum,

We are pleased to inform you that your manuscript 'The hierarchical organization of autocatalytic reaction networks and its relevance to origin of life' has been provisionally accepted for publication in PLOS Computational Biology.

Best regards,

Philippe Nghe

Guest Editor

PLOS Computational Biology

James O'Dwyer

Section Editor

PLOS Computational Biology

Reviewer's Responses to Questions

**Comments to the Authors:**

Reviewer #1: Thanks for the detailed responses to my previous questions.

**Have the authors made all data and (if applicable) computational code underlying the findings in their manuscript fully available?**

Reviewer #1: None

PLOS authors have the option to publish the peer review history of their article (what does this mean?). If published, this will include your full peer review and any attached files.

Reviewer #1: No

---

## [Editor Report · Acceptance letter]

1 Sep 2022

PCOMPBIOL-D-22-00678R1 

The hierarchical organization of autocatalytic reaction networks and its relevance to the origin of life

Dear Dr Baum,

I am pleased to inform you that your manuscript has been formally accepted for publication in PLOS Computational Biology. Your manuscript is now with our production department and you will be notified of the publication date in due course.

With kind regards,

Livia Horvath
